

# Climate change will increase potential hydropower production in six Arctic Council member countries based on probabilistic hydrological projections

Elena Shevnina[1], Karoliina Pilli-Sihvola[1], Riina Haavisto[1], Timo Vihma[1], Andrey Silaev[2]

[1] Finnish Meteorological Institute, Erik Palménin aukio 1, FI-00560, Helsinki, Finland
[2] National Research University Higher School of Economics, 25/12, Bolshaya Pecherskaya Street, 603155, Nizhny Novgorod, Russia

*Correspondence to*: Elena Shevnina (elena.shevnina@fmi.fi)

**Abstract.** Potential hydropower production for 2020-2050 is calculated for 173 catchments located over the territories of
Finland, Sweden, Norway, the Russian Federation, Canada and the United States. The results are based on hydrological river runoff projections assessed together with their exceedance probabilities. The annual runoff rate of particular exceedance probability was modelled with the Pearson type 3 distribution from three parameters (mean values, coefficient of variation and coefficient of skewness) simulated by the probabilistic hydrological MARcov Chain System (MARCS) model. The probabilistic projections of annual runoff were simulated from outputs of four global climate models under three
Representative Concentration Pathways (RCP2.6, RCP4.5 and RCP8.5). The future potential hydropower production was evaluated based on annual runoff of low and high exceedance probabilities, and then aggregated at a country level. Under forcing from climate models that project a large increase in precipitation (CaEMS2 and MPI-EMS-LM), the expected potential hydropower production in the six countries increased by 14.0 to 18.0 % according to the projected values of annual runoff rate on exceedance probabilities of 10 and 90 %. This increase in water resources allows for 10–15 % more hydropower energy generation by
rivers located in Russia, Finland, Norway, and Sweden. For the USA and Canada, the potential hydropower production is projected to increases by 4.0–9.0 %. Under forcing from climate models that project a smaller increase in precipitation (HadGEM2-ES and INMCM4), the increase of potential hydropower production by 2050 was predicted to be 2.1–8.4% over the six countries considered.

## 1 Introduction

Water has always been a key natural resource in energy production. Nowadays, hydropower plant operators use deterministic
and probabilistic hydrological forecasts on stream flow water runoff (Schwanenberg et al., 2014; Xu et al., 2014; Domínguez and Rivera, 2010; Shevnina, 2001) to optimize rules of surface runoff regulation and energy production in hydropower plants (Tucci et al., 2008), and to prevent losses due to extreme runoff events. Engineering hydrology introduces the extreme runoff events in terms of exceedance probability as multi-year stream flow runoff values of low or high occurrences (van Gelder et al., 2006; Wilson, 1990; Rozhdestvenskiy and Chebotarev, 1984). In hydropower industry, the extreme runoff events result to





losses caused by water spills due to flooding, or lead to interruptions in operation of hydropower plants due to water
shortages. On the long-term, the hydropower industry is sensitive to changes in water resources available for energy
production, and changing patterns in the extreme runoff events (Döll and Schmied, 2012; Madsen et al., 2013; Milly at al.,
2008).

Long-term hydrological predictions have become important for hydropower producers due to observed trends in the surface

river runoff, which are related to climatological changes in air temperature and precipitation. According to the
Intergovernmental Panel on Climate Change (IPCC) Fifth Assessment Report (AR5; IPCC, 2013), the anomaly in annual
precipitation rate (mm year$^{-1}$) over the Arctic has about 30-60 %, and precipitation generally tends to increase during the past
decades as shown in Fig. 2.28 of the IPCC report (IPCC, 2013). Trends in annual mean precipitation in the Arctic and northern
mid-latitudes show a large spatial variation, and are sensitive to the time period addressed. Further, inaccuracies in snow fall

observations (Alexandrov et al., 2005) contribute to the inaccuracy of precipitation trends. Observations suggest a general increase
in annual precipitation in circumpolar high-latitude regions over recent decades (Vihma et al., 2016; Roshydromet, 2014; Hartmann
et al., 2013;). Increases have been reported, among others, for northern Canada and a large part of Russia. In the mid-latitude
regions of the United States, the long-lasting drought generated by the decrease in annual precipitation during last decade
(Barnstone and Lyon, 2016). Winter and summer precipitation have decreased in the headwater parts of the Mackenzie River basin

(Yip et al., 2012), and summer and autumn precipitation have decreased in central Eurasia (Bogdanova et al., 2010). The general
increase in precipitation is due to the global warming trend (Hartmann et al., 2013).

In Northern Hemisphere land areas, the annual mean precipitation typically increases towards south, as the water-holding capacity
of air depends on a temperature. Coastal regions, in particularly east of oceans, represent the largest deviations from this north-
south trend. Precipitation amounts are anomalously large in the west coast of Canada, south-west coast of Alaska, north-west coast

of the USA, and along the Norwegian coast (Vihma et al., 2016). Lowest precipitation amounts are observed in northern Siberia
and Canadian Arctic archipelago.

Rosmann et al. (2016) revise the observed global river runoff time-series, and find that the highest number of statistically
significant trends are detected in the multi-year time series of an annual stream flow runoff. The trends in observed surface
river runoff motivate hydrologists to revise the basic assumption behind the long-term water resources and extreme runoff

predictions, because historical observations do not provide information on future risks connected to water resources and
stream flow runoff extremes (Tananaev et al., 2016; Madsen et al., 2013; Milly at al., 2008). The annual stream flow runoff
serves water resource for the hydropower industry, which is an important element on an energy production in the Arctic
Council member countries especially on the Nordic countries (FI, NO, SE) (Norden, 2018). This study focused on the annual
surface river runoff, which was considered as a random variable  analysed on the basis of multi-year time series of observed

annual river discharges s or stream flow rates.

To express a general relationship between water use and water resources available, numerous use-to-resource ratios are
suggested (Brauman et al. 2016; Tidwell et al., 2012). Examples include the Water Stress Index, defined as the ratio of
available river runoff to population in a basin (Falkenmark et al., 1989) and the water supply stress index, which considers



regional trends in both water supply and demand (e.g., Averyt et al., 2013). Roy et al (2005) take a similar approach except
that their metric is constructed as the ratio of water withdrawal to effective precipitation. Other metrics are based on multiple
criteria that are aggregated and related to some threshold of water availability/sustainable development (Xu and Wu, 2017).
For hydropower, a potential hydropower production (PHP) is a traditional indicator, usually evaluated from the mean values
of the annual river discharges (Parkinson and Djilali, 2015; Hamidudu and Killingtveit, 2012). However, focusing solely on
the mean value is not sufficient, as future risks on water resources are posed by changes in the temporal dynamics of river
discharge (Döll and Schmied, 2012) and, on hydropower production, particularly by changes in the runoff extremes
(Blackshear et al., 2011). Water engineering defines the stream flow runoff extremes as values of low/high exceedance
probabilities, which are estimated from tails of exceedance probability curves (EPCs) of river runoff. The exceedance
probability indicates a likelihood that a particular runoff value will be exceeded. The probabilistic hydrological projection
gives an opportunity to increase our understanding of future risks on hydropower generation through evaluation of the
potential energy production expressed in terms of probability.

While many studies are directed to the evaluation of PHP, they are only based on estimates of the future mean value for the
annual stream flow runoff. For example, Madany and Lund (2010) assess the projected mean values for annual river runoff
from a proportion to historical mean values for California, and use these as inputs in an optimization model to define optimal
regimes of future hydropower generation (Madany and Lund, 2009). Lehner et al. (2005) use the global-scale WaterGAP
model (Alcamo et al., 2003), taking into account both climate and socio-economic changes, to evaluate PHP for Europe. A
hydrological rainfall-runoff model (Döll et al., 2003) is used to simulate the mean values of annual river runoff based on
future precipitation and air temperature available as climate model outputs. Hamududu and Killingtveit (2012) provide a
state-of-the-art assessment for global installed hydropower capacity, and focus on the future changes in PHP based on
projections from 12 global climate models (Milly et al., 2005). However, the results of these and many other studies do not
provide estimates on how the future extreme variability of water resources may affect potential hydropower production.
In this study, the potential hydropower production for the river runoff values of low/high exceedance probabilities are evaluated
over 173 catchments located in six out of the eight member countries of the Arctic Council for 2020–2050: Finland (FI), Sweden
(SE), Norway (NO), Russian Federation (RF), Canada (CA) and the United States (US). The share of hydropower production out
of the total electricity generation in each country, except for the US and RF, is 20% or over (World Development Indicators on
http://wdi.worldbank.org/table/3.7). In Norway, it is over 95%, in Canada almost 60%, in Sweden 40% and in Finland 20%. The
two other Arctic Council member countries, Iceland and the Kingdom of Denmark also have major freshwater resources and
hydropower generation. However, they were excluded from the analysis as no data for large (1,000-50,000 km$^2$) watersheds located
in Iceland and Greenland were found. The data and methodological details are described in Section 2. The results are shown in
Section 3, and Section 4 discusses the limitations of our study and future research needs. Among the countries considered, the three
Nordic countries (FI, NO, SE) do not have plans to build new hydropower capacity in the near future. The Russian Federation has
some facilities planned, and the USA and Canada are currently planning and building major hydropower facilities



(http://atlas.freshwaterbiodiversity.eu/atlasApp/full/index.html?map=3.4.3-global-hydropower-dams) based on Zarfl et al., (2015). This also affects how the information of future climate change can be incorporated in decision making in the different phases of hydropower plant management, including planning, upgrading and operation of the plants. This is briefly discussed in Section 5, which concludes the paper.

## 2 Method and Data

### 2.1 Method

The modeling approach used in this study consists of three components: an input (model forcing), the model (a conceptual abstraction or mathematic equation) and an output (simulation results). Figure 1 shows a schematic representation of the flow of the models and their outputs as well as the forcing data used with the framework of this study. The study method includes climatological, hydrological and economical models, which form the chain from climate projections via water resource projections to economic indicators and indexes. The historical yearly time series of the annual river discharges serve as the input to estimate the mean, coefficient of variation (CV) and coefficient of skewness (CS) of annual runoff. These time series were observed on 326 sites of the national hydrological networks of the six Arctic Council member countries. The yearly time series of annual precipitation amount were extracted from open sources and used to estimate historical and protected mean values to set-up and force the probabilistic hydrological MARkov Chain System model MARCS (the middle block on Hydrology column in Figure 1).

The output of the MARCS model is the exceedance probability curve of annual river runoff (expressed in units of stream flow rate or of water discharge). In this study, the values of runoff of low and high exceedance probabilities were used to calculate the potential hydropower production for the projected period (the upper block of the Hydropower column on Figure 1). The simple relationship between energy production and water discharge on any site was used as the economic model (see details further).

### 2.1.1 Climatology

In this study, the historical observations on annual precipitation amount and river water discharge were used in estimations of the non-central moments, the first (mean value), second and third moments. The estimates of all three non-central moments were used to estimate mean values, coefficient of variation and coefficient of skewness of the river water discharges at gauging sites to set-up the hydrological MARCS model. The mean values were only estimated from the time series of annual precipitation amount for the historical period as well as for the projected period (2020–2050). The outputs from four global climate models under three climate change scenarios were used in this study (see further details in Section 2.2).



### 2.1.2 Hydrological model

An Advanced Frequency Analysis (AFA) suggested by Kovalenko (1993) was applied to evaluate the probabilistic projections of annual river runoff. The AFA method combines traditional modeling methods (Alcamo et al., 2003) with frequency analysis methods (van Gelder et al., 2006), and it is a part of the Fokker-Plank-Kolmogorov equation approach (Rosmann and Domínguez, 2017; Domínguez and Rivera, 2010; Kovalenko, 1993). The main idea of the AFA method is to simulate statistical estimators of annual river runoff (mean, variation and skewness) from the mean annual precipitation (Shevnina et al., 2017; Kovalenko et al., 2010; Kovalenko, 1993; Pugachev et al., 1974). In this study, the MARCS model (Shevnina and Krasikov, 2018; Shevnina and Gaidukova, 2017; Shevnina, 2015) was used to simulate the mean, CV and CS of annual runoff rate (ARR) for the selected gauging sites (see black dots on Figure 2).

To set up the MARCS model, the historical ARR time series (Table 1) were analysed for trends, stability of variance and mean by applying the Spearman's Rank-Correlation Test (SRC), the Fisher's *F*-test (FR) and Student's *t*-test (ST), as suggested in Dahmen and Hall (1990). The FR and ST values were evaluated using sub-series divided at a fixed year (1975), and the values of SRC were evaluated by a "floating point" technique (Shevnina et al., 2017). The same technique was applied to define the reference period, and its length varied for the selected gauging sites. The reference period covers a period in the past without statistically significant trends in the observed time series of annual runoff. In our study, the reference period was specific for each catchment. For the reference period, the estimates of the three first non-central statistical moments were evaluated from the historical time series of ARR with the method of moments (Rozhdestvenskiy and Chebotarev, 1984). The projected period chosen is 2020-2050 for all gauging sites.

To parametrize and force the MARCS model, the mean values of annual precipitation were evaluated for both the reference and projected period with gridded climatological datasets. Thus, the means of annual precipitation were calculated at a grid node nearest to a centroid of a catchment outlining a gauging site. The projected mean values of annual precipitation were evaluated from outputs of four global climate models and corrected with the delta method (Fowler et al., 2007).

The MARCS model simulates three non-central moments of ARR by allowing the calculation of the Pearson type 3 (Pt3) probability distribution parameters used in water engineering (Koutrouvelis and Canavos, 1999; Rozhdestvenskiy and Chebotarev, 1974; Matalas and Wallis, 1973). In this study, only two non-central moments of ARR were simulated for each projection of the future climate, which were then used to calculate CV. The projected CS was evaluated with a CS/CV ratio considered to be constant for the reference and projected periods (Shevnina et al., 2017; Kovalenko et al., 2010).

To evaluate the projected annual runoff rate of a low/high exceedance probability, a look-up table (Salvosa, 1930) was used. First, the ordinates of the Pearson type 3 (Pt3) distribution were estimated for each river basin, and then further applied in calculation of the annual runoff rate of high/low exceedance probability. The results were aggregated at a country level to estimate the changes on the PHP.



### 2.1.3 Potential hydropower production

Energy generation is among the most important indicators of a hydropower plant efficiency. The PHP (in Watts) provides an estimate of the energy production. Hydropower engineering handbooks (e.g., Obrezkov, 1988) provide a simple relationship between PHP and water resources, formulated as follows:

$$PHP = \rho \zeta H Q \quad , \tag{1}$$

where $Q$ is the mean annual water discharge (m$^3$ s$^{-1}$) at a plant site, $H$ is a plant-specific hydraulic head (m), and $\rho$ and $\zeta$ are water density (kg m$^{-3}$) and gravity acceleration (m s$^{-2}$), respectively.

Eq. 1 presents a general relation between energy production and water resource (annual runoff) in production facilities. The PHP depends on the available annual river runoff (a resource) and site-specific hydropower plant equipment (a technology), and
allows estimation of an average hydropower production in global and regional scales for the near future (Parkinson and Djilali, 2015; Hamududu and Killingtveit, 2012) in assumption that the technology will not change. In our study, PHP was considered a random variable linearly related to the annual runoff, which is also defined as a random variable ( $PHP_p = \rho \xi H Q_p$ ). The future relative change in PHP (dPHP) was evaluated on the basis of the mean annual runoff and the upper (p=10%, high flow) and lower (p=90%, low flow) tails of ARR with an assumption that the technology in hydropower production remains same during the
next 30 year. Thus, dPHP is only proportional to changes in the mean ARR as well as ARR of low and high probability of exceedance. The MARCS model outputs were used to evaluate the changes of PHP for the 173 catchments located over the six Arctic Council member countries for the period of 2020–2050.

### 2.2 Data

To set-up, parametrize and force the MARCS model for the selected catchments, the observed yearly time series of water discharge
and precipitation as well as of the precipitation for the projected period were collected from open-source datasets. The historical time series of water discharge were extracted from the Global Runoff Data Center (GRDC ), as monthly series for 326 gauging sites. The sites are located in the territories of Finland, Sweden, Norway, Russia, Canada and United States (Table 1). The observations cover a period of 1863–2015.

The average catchment area of the selected sites is 12,000 km$^2$. The length of the time series varies from 30 to 151 years, and
exceeds 80 years on average. The Eurasian part of the Arctic is presented by 147 gauges, and 179 gauges are distributed over North America. The annual discharges were calculated from monthly discharges, and then expressed as ARR (stream flow per unit basin area, mm yr$^{-1}$). If the ARR time series was shorter than 35 years or included gaps of more than 10 years, the time series was excluded from further study. A total of 219 gauging sites fulfilled the required criteria on time series length and gaps. The inhomogeneity in the observed time series of ARR was detected by the Student t-test (ST), the Fisher's F-test
(FR), and the Spearman Rank-Correlation Test (SRC).




The time series of annual precipitation rate (mm yr$^{-1}$) for 1900 to 2010 were extracted from the UDel_AirT_Precip dataset provided by NOAA/OAR/ESRL PSD, Boulder, Colorado, USA, via their web site at http://www.esrl.noaa.gov/psd/. The dataset of the Coupled Model Inter-comparison Project 5, CMIP5 (Taylor et al., 2012) was used to force the MARCS model. The model forcing was evaluated for 2020–2050 from the outputs of the four climate models HadGEM2-ES (Collins et al., 2011), INMCM4
(Volodin et al., 2010), CaEMS2 (Chylek et al., 2011) and MPI-ES-LR (Giorgetta et al., 2013) under three Representative Concentration Pathways (RCP2.6, RCP4.5 and RCP8.5).

## 3 Results

### 3.1 The model set-up: reference period

Figure 2 shows the gauges with the observed time series of river water discharge. The ARR time series were analysed for
trends and the stability of mean and variance. Statistically significant trends were detected in 25 % of the ARR time series, observed mostly south of the Arctic Circle. These trends may be related to artificial water regulations or natural factors, i.e. climate change. We do not explore the reasons for the trends here, and simply exclude the non-homogeneous ARR time series from the analysis.

For the remaining 208 catchments, the inverse of runoff coefficient (IRC) was calculated by dividing the mean annual
precipitation rate by the mean annual runoff rate. The IRC generally reflects watersheds' physiography, and gives estimates on portions of evaporation and surface water runoff in total precipitation (Kovalenko, 2011; Sokolovskiy, 1968). The closer the IRC value is to 1.0, the larger is the portion of surface runoff in precipitation. Outliers, i.e., the catchments with the IRC values less than 1.0 or more than 7.0 (the red dots in Figure 3a) were also excluded from the further study, since such values must be caused by some specific factors. For instance, if IRC < 1.0  it generally means that the surface runoff exceeds
incoming precipitation. This case may occur in a river basins with multi-year artificial regulation. When IRC > 7.0 the portion of surface runoff in precipitation is very small, and almost all water is evaporated (Kovalenko, 2011).

To set-up the MARCS model, the mean values of annual precipitation rate (PRE, Figure 3b), first and second non-central moments (M1 and M2, Figure 3 c and d) as well as the third non-central moment (not shown) were estimated for the reference period. Then the estimates of the three moments of ARR were used to calculate the CS/CV ratio (not shown)
applying the basic parameterization scheme (Shevnina, 2015; Kovalenko, 1993). This parameterization provides over 80 % of successful hindcasts made on historical data (Shevnina et al., 2017; Kovalenko, 1993).

Most surface water resources available for hydropower generation in the case study area are located over the North American coast, in Fennoscandia and western part of the Russian Federation. In the coastal areas, surface water resources mostly depend on precipitation; the role of evaporation is smaller because runoff is enhanced by orography and, in summer (when
precipitation amounts are generally largest), surface temperatures are usually relatively low compared with inland regions (Jin, 2004). Figure 2d shows that the highest mean values of ARR are obtained on coastal areas with maritime climate, where the mean values of annual precipitation rate are also high (Figure 2b).





The estimate for the second non-central moment (Figure 2c) also shows the maximum values in these regions, where the range of the inter-annual variability of water resources is higher than for the inland areas. This is probably related to large-scale atmospheric circulation. The coastal zones of North America and Norway are the regions where the annual mean precipitation reaches its maxima. Much of the precipitation is related to advection of moist marine air masses, carried by transient cyclones, to the coastal zone where orographic lifting results in precipitation (e.g., Jakobson and Vihma, 2010). Hence, inter-annual variations in the cyclone activity and tracks result in large variations in precipitation and further in water resources. Inter-annual variations in the cyclone activity and tracks are large also in inland areas but, as the magnitude of precipitation is smaller, their effect on the second non-central moment remains smaller.

Table 2 provides a summary of the MARCS model setup over the 173 watersheds studied. The results are grouped by country to show the basic statistics for IRC, M1, PRE, CV and CS/CV of ARR evaluated from the historical time series. For the catchments located on Norway, the IRC values are about 1.0. This generally means that annual precipitation mostly feeds surface runoff on rivers regulated by numerous hydropower plants located over the catchments. The losses from evaporation are minor. The IRC maxima were obtained for the rivers located in the United States, where the role of evaporation is high, especially for inland catchments. The IRC values are approximately similar for the watersheds located in Finland, Sweden, Russia and Canada.

### 3.2 The model forcing: projected period 2020–2050

The mean values of annual precipitation rate over the period 2020-2050 ($PRE_{2050}$, mm yr$^{-1}$) were obtained from the outputs of HadGEM2-ES, CaEMS2, INMCM4 and MPI-ES-LR climate models under the RCP2.6, RCP4.5 and RCP8.5 scenarios. Table 2 shows the summary for the expected changes in $PRE_{2050}$ averaged over the catchments considered. The values of $PRE_{2050}$ slightly vary over the RCP scenarios, and different climate models suggest different ranges for the future $PRE_{2050}$: the HadGEM2-ES and INMCM4 predict a small decrease on annual precipitation rate up to 5.2% (Table 3, HAD-26), whereas the CaEMS2 and MPI-ES-LR models propose an increase of the $PRE_{2050}$ up to 4.8 % (Table 3, CAD-26). Figure 4 shows the relative changes in the mean annual precipitation rate ($dPRE_{2050}$, %) calculated by dividing the difference between projected ($PRE_{2050}$) and reference (PRE) values by the reference (PRE) value. As the INMCM4 model gives results very similar to those from the HadGEM2-ES, we do not present them in Figure 3 and further figures but only in the tables.

Based on the range of the predicted $dPRE_{2050}$, the climate models used can be divided in two types. The CaEMS2 and MPI-ES-LR models suggest twice stronger increase of annual precipitation rate than the HadGEM2-ES and INMCM4 models (Table 3, Figure 4). The $dPRE_{2050}$ values for more than 10% were predicted for a number of catchments (dots with yellow to red colors in Fig. 4) with the highest changes up to 60 % in Eastern Siberia. In contrast, the HadGEM2-ES and INMCM4 models produced no changes or a slight decrease of annual precipitation rate for most of the catchments. In this case, the $dPRE_{2050}$ values varied within 10 %, (blue and green dots in Figure 4).



### 3.3 The model output: annual runoff

The projected values for two non-central moments of ARR were simulated for each climate model and RCP scenario combination, and two parameters of the Pt3 distribution, denoted as $M1_{2050}$ and $CV_{2050}$, were calculated following Rozhdestvenskiy and Chebotarev (1974) and Elderton and Johnson (1969). Then, the relative changes $dM1_{2050}$ ($dM1_{2050} = (M1_{2050} − M1)/M1$, %) and $dCV_{2050}$ (($CV_{2050} − CV)/CV$, %) were used to classify the catchments into six classes based on thresholds. Kovalenko (1993) suggested a classification based on modelling errors inherent to simulated parameters of Pt3

distribution, and defined the thresholds to be considered significant for the mean value and CV of ARR. In our study, the thresholds of $−10\% < dM1_{2050} < 10\%$ and $−20\% < dCV_{2050} < 20\%$ were considered significant according to Kovalenko (1993). The projected $CV_{2050}$ vary a little compared to the CV reference values under all climate model and RCP scenario combinations considered. The relative changes $dCV_{2050}$ were within the thresholds $−20\% < dCV_{2050} < 20\%$, thus they were neglected during watersheds selection.

Based on the outputs of HadGEM2-ES and INMCM4 models, the MARCS model predicted lowest $dM1_{2050}$ values in a majority of the catchments. However, $dM_{2050}$ exceeding 10 % was projected only for a fP26 (Figure 5, b). The changes in the projected mean values of ARR were estimated to be in a similar range under the CaEMS2 and MPI-ESM-LM models. This range is about twice more than for the $dM1_{2050}$ values calculated from the forcing by the HadGEM2-ES and INMCM4 models (Figure 5 a,c,d,f). The forcing by the MPI-EMS-LM under the RCP85 results on a highest increase of mean values of

ARR; more than 30-40% increase in catchments located in the Siberian territories of Russia as well as the Yukon territory of Canada. In the Nordic countries, the increase of the mean values of ARR was projected to be up to 15–20% in most of the catchments with the MPI-EMS-LM and CaEMS2 climate models (Figure 5 a,c,d,f; Table 4).

The values of $M1_{2050}$ and $CV_{2050}$ were evaluated on average over all catchments located in a particular country to provide a "country level" analysis based on the probabilistic projections of ARR. The values of $M1_{2050}$ and $dCV_{2050}$ were averaged over

the catchments selected, and then classified into two cases: "wet" forcing and "dry" forcing. The wet forcing summarized the results under the CaEMS2 and MPI-EMS-LM models whereas the dry forcing provided the estimates under the HadGEM2-ES and INMCM4 models (Table 4).

Comparing the reference values of ARR reveals that the projected values of $M1_{2050}$ increase by 5.0–15.0 % under the wet forcing and there is a significant increase in water resources in the six Arctic Council member countries. Under the "dry"

climate forcing, the increase varied between 1.5 and 7.0 %, and thus is considered to be insignificant. Also the $dCV_{2050}$ under both wet and dry forcing is found to be insignificant (Table 5).

To evaluate the projected annual runoff rate ($ARR_{2050}$) of low and high exceedance probability, the Pearson type 3 (Pt3) distribution was applied (Koutrouvelis and Canavos, 1999; Rozhdestvenskiy and Chebotarev, 1974; Matalas and Wallis, 1974). Following the look-up table applied in the engineering hydrology (Salvosa, 1930), the ordinates of Pt3 distribution

were evaluated for two exceedance probabilities (k10 and k90 in Table 6) based on the projected $CV_{2050}$ and CS/CV ratio averaged on a country level (Table 5). These ordinates were further used in calculation of the annual runoff rate of high





(ARR10$_{2050}$) and low (ARR90$_{2050}$) exceedance probability. ew watersheds located in the west coast of Canada, the east coast of the United States and eastern parts of European Russia. Among the RCP scenarios, the highest values of dM$_{2050}$ were predicted under RCP26 (Figure 5, b). The changes in the projected mean values of ARR were estimated to be in a similar range under the CaEMS2 and MPI-ESM-LM models. This range is about twice more than for the dM1$_{2050}$ values calculated from the forcing by the HadGEM2-ES and INMCM4 models (Figure 5 a,c,d,f). The forcing by the MPI-EMS-LM under the RCP85 results on a highest increase of mean values of ARR; more than 30-40% increase in catchments located in the Siberian territories of Russia as well as the Yukon territory of Canada. In the Nordic countries, the increase of the mean values of ARR was projected to be up to 15–20% in most of the catchments with the MPI-EMS-LM and CaEMS2 climate models (Figure 5 a,c,d,f; Table 4).

The values of M1$_{2050}$ and CV$_{2050}$ were evaluated on average over all catchments located in a particular country to provide a "country level" analysis based on the probabilistic projections of ARR. The values of M1$_{2050}$ and dCV$_{2050}$ were averaged over the catchments selected, and then classified into two cases: "wet" forcing and "dry" forcing. The wet forcing summarized the results under the CaEMS2 and MPI-EMS-LM models whereas the dry forcing provided the estimates under the HadGEM2-ES and INMCM4 models (Table 4).

Comparing the reference values of ARR reveals that the projected values of M1$_{2050}$ increase by 5.0–15.0 % under the wet forcing and there is a significant increase in water resources in the six Arctic Council member countries. Under the "dry" climate forcing, the increase varied between 1.5 and 7.0 %, and thus is considered to be insignificant. Also the dCV$_{2050}$ under both wet and dry forcing is found to be insignificant (Table 5).

To evaluate the projected annual runoff rate (ARR$_{2050}$) of low and high exceedance probability, the Pearson type 3 (Pt3) distribution was applied (Koutrouvelis and Canavos, 1999; Rozhdestvenskiy and Chebotarev, 1974; Matalas and Wallis, 1974). Following the look-up table applied in the engineering hydrology (Salvosa, 1930), the ordinates of Pt3 distribution were evaluated for two exceedance probabilities (k10 and k90 in Table 6) based on the projected CV$_{2050}$ and CS/CV ratio averaged on a country level (Table 5). These ordinates were further used in calculation of the annual runoff rate of high (ARR10$_{2050}$) and low (ARR90$_{2050}$) exceedance probability.

### 3.4 Potential hydropower production

The changes in the potential hydropower production (dPHP) were estimated to be simply proportional to the changes of ARR (dARR$_{2050}$). The dARR$_{2050}$ was evaluated on the basis of the mean values of ARR and the values of low and high exceedance probabilities (dARR10$_{2050}$ and dARR90$_{2050}$) under the wet and dry climate forcing. It was assumed that the technology in hydropower production (such as hydraulic heads and efficiency of water turbines) remains same during the next 30 year. Thus, it was assumed that dPHP only depended on dARR, dARR10$_{2050}$ and dARR90$_{2050}$ (Figure 6).

Over the six Arctic Council member countries, the expected potential hydropower production (PHP$_{2050}$) increased by 14–18 % according to the projected values of ARR10$_{2050}$/ARR90$_{2050}$ under the wet climate forcing. This increase in water resources allows for 10–15 % increase in hydropower generation in Russia and Fennoscandia (FI, NO, SE) according to the forcing by the CaEMS2



and MPI-EMS-LM climate models. For the United States and Canada, the expected potential hydropower production increased by 4.0–9.0 %. Under the forcing by the HadGEM2-ES and INMCM4 climate models (dry forcing), the $dPHP_{2050}$ was predicted to be 2.1–8.4% over the six countries considered.

## 4 Discussion

Several features of the current version of the MARCS model should be analyzed more carefully to understand its capacity to 320 provide reliable projections of future runoff and potential hydropower production. Climate models simulate meteorological variables in a grid, and the values are assumed to be representative of each grid box. As the global climate models have a coarse spatial resolution, this assumption becomes critical for physically based and spatially distributed hydrological models (Hostetler, 1994). In this case, regional-scale climatology usually serves as the meteorological forcing for the physically based basin-scale hydrological models (Xu, 1999). The sensitivity of the MARCS model to spatial resolution of forcing by global and regional 325 climate models is a topic for a further study.

One major benefit of using the MARCS model is that it simulates the mean, CV and CS of annual runoff, allowing the calculation of the parameters of exceedance probability curves (*i.e.* Pt3) with little computational burden compared to physically based hydrological models. The MARCS model produces the probability distribution of runoff from statistics of meteorological variables, and it requires less resources. Thus, the exceedance probability curves of AAR for a number of catchments are simulated from the 330 output of a single climate model without special requirements to computational facilities. It makes the MARCS model useful in regional-scale estimations on extreme floods, droughts and the risks related to their occurrence (Shevnina et al., 2017; Shevnina, 2015; Shevnina, 2014). In our study, the MARCS model was forced by a mean annual precipitation rate of a single climate projection; however, it can be also applied on ensemble of climate projections (Tebaldi and Knutti, 2007). Then, the probabilistic projection of annual runoff allows producing "two-dimensional" version of a probabilistic hydrological projection. In this case, the 335 first dimension indicates an ensemble statistics of annual runoff rate of a particular exceedance probability (to be presented by the second dimension). Ensemble modeling is a topic for a continuation study.

In our data on ARR time series, a trend and non-homogeneity were detected in about 25 % of the time series observed on the watersheds located mostly south of 60° N. This is in line with the results in Rosmann et al. (2016), who detected trends in observed time series of daily, monthly and annual precipitation, air temperature and river discharge; the highest number of trends were 340 detected in annual river discharge. Déry et al. (2009) detected trends in CV of annual discharges on over 30 % of time series of water discharges observed by Canadian gauges. The authors conclude that this fact "provides observational evidence of an intensifying hydrological cycle in northern Canada, consistent with other regions of the pan-Arctic domain". Increasing of annual river discharge in Eurasian rivers of is also reported by other studies (Tananaev et al., 2016; Shiklomanov et al., 2006; Peterson et al., 2002), and our estimates are generally in consistence with others.





Generally, all RCP scenarios used in the study predicted similar changes in the mean annual precipitation. The four global models predicted an increase in PRE, ranging from 10% for the catchments located south of 60° N to 60 % for the Eastern Siberian watersheds located north of 60° N. Similar conclusions on the future increase of precipitation over land surface in the Polar regions are obtained previously by a number of authors (Kusunoki et al., 2015; Prowse et al., 2015; Rawlins et al., 2010; Pavelsky and Smith, 2006).  The Arctic amplification of climate warming is projected to be associated with intensification of the water cycle,

with increases in precipitation, evaporation and moisture transport from mid-latitudes (Lique et al., 2016; Vihma et al., 2016; Bring et al., 2016; Taylor et al., 2013). Hence, to assess a role of evaporation, the projected values of air temperature need to be included among forcing variables of the MARCS model. This possibility is discussed in Kovalenko et al. (2010), and some studies report improvement of the results on estimations based on historical hindcasts (Shevnina, 2015; Kovalenko et al., 2006).

We note that this study has only addressed projections for the period 2020-2050. Due to the major role of inter-annual and decadal

variability, the projected precipitation trends for this period depend on each climate model's realization of these variations. Considering projected changes by the end of the 21$^{st}$ century, the relative importance of inter-annual and decadal variations becomes smaller, and the CMIP5 model projections show more robust trends of increasing precipitation over the Arctic and mid-latitudes (Collins et al., 2013; Lique et al., 2016).

Recently, a number of studies have addressed the changes in annual runoff mean values (Prowse et al., 2015; Bengtsson et al.,

2011; Lehner et al., 2005 ) but do not provide estimates on CV or ARR of 10% and 90% exceedance probability. Thus, we can compare our results only with the estimates for the mean values of ARR provided in the previous findings. In particular, Prowse et al. (2015) concludes that "the higher-latitude terrestrial areas of the Arctic are generally becoming more "water rich". The highest increase in mean values of ARR of more than 30–40% were projected for the catchments in Siberia in the Russian Federation and the Yukon territory in Canada (Döll and Schmidt,2012; Frigon et al., 2010). However, only this study evaluated the

ARR values of 10 and 90 % exceedance probability in additional to the mean AAR under each climate projection and climate model output for 173 catchments located over six Arctic member countries.

While numerous studies are addressed to risks for the hydropower production due to climate changes (Parkinson and Djilali, 2015; Hamududu and Killingtveit, 2012; Bengtsson et al., 2011; Seljom et al., 2011), they are mostly based on the future mean values of the annual runoff. In our study, the ARR values of 10 and 90 % exceedance probability allow evaluating the changes in the

potential hydropower production of low and hight probability, thus the risks can be evaluated quantitatively, not only qualitative. In this study, we only considered relative changes in the potential hydropower production since the results of the hydrological modeling were aggregated on a country level. It means, that the parameters of the Pearson type 3 distribution simulated by the MARSC model were averaged over the watersheds located within each country. However, the future potential hydropower production can be analyzed in terms of probability also on "basin-by-basin", or even for a site coinciding with an existing

hydropower plant. In this case, it is possible to include specific plant details such as hydraulic head and turbine efficiency, and calculate future exceedance probability for potential hydropower production in absolute values.

**5 Conclusions**



Historical time series are losing their role as the only data source for long-term planning of hydropower production. Changing climate is altering the river surface runoff, which is the key hydrological parameter for hydropower production. Several studies have analyzed how global and regional mean annual runoff will change in the future due to climate change. We extended these analyses by estimating the annual runoff of low and high exceedance probability. Long-term probabilistic hydrological projections allowed us to estimate the runoff values together with their exceedance probabilities from the Pearson type 3 distribution, based on the simulations of three parameters (mean value, coefficient of variation and coefficient of skewness). We simulated these three parameters for the period 2020-2050 applying the MARCS model and outputs of four global climate models under three RCP scenarios.

Our analysis indicates that the future potential hydropower production in the territories of Canada, Finland, Norway, Russian Federation, Sweden, and the USA will generally increase regardless of differences between the climate forcing scenarios. However, the magnitude of the increase depends heavily on the forcing. Based on the two models that resulted in so-called wet forcing, climate change will allow for a 10–15% increase in the potential hydropower generation in the Russian Federation and Fennoscandia, whereas in the USA and Canada, the increase is 4.0– 9.0%. Based on the two models that resulted in so-called dry forcing, climate change will allow for a 2.1–8.4% increase in the potential hydropower generation in all the six countries.

The probabilistic form of forecasts provides a solid basis for decision-making in cost-lost situations (Mylne, 2002; Murphy, 1977, 1976). Among the countries addressed in this study, only the United States and Canada have major plans for new hydropower capacity in their territory. Hence, they have the highest potential in incorporating our results in the planning and design of the new plants. However, the uncertainty in the results requires location-specific further analyses on how climate change will affect hydropower production in each catchment. In the other countries addressed, potential increase in hydropower production should be considered during upgrades of the hydropower plants and optimizations of multi-year regulation rules for the plants. In general, the results show that the impacts of climate change should be assessed in detail when hydropower plants are planned and designed.

**Data availability**

The data files for the MARCS model set up (MARCS_setup.csv), forcing (MARCS_forsing.csv) and output (MARCS_results.csv) were uploaded as supplements to this paper as well as the code behind the Tables 2–6 (Tables_Shevninaetal2018.py).

**Author contribution**

E. Shevnina designed the hydrological model and code, collected the data and performed the simulations. K. Pilli-Sihvola and R. Haavisto analysed recent data on the hydropower production in six Arctic member countries. T. Vihma revised the recent trends and the future projections in precipitation and over the Arctic, A. Silaev contributed to define general relationship between water resource and hydropower production. E. Shevnina prepared the manuscript with contributions from all co-authors.



**Competing interests**

The authors declare that they have no conflict of interests.

**Acknowledgements**

The study is supported by the Academy of Finland (contract 283101).

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





**Table 1. Overview of the runoff dataset used in the study**

| Country | | Number of gauges | Catchment area, km$^2$ | | Length of time series, year | |
|---|---|---|---|---|---|---|
| | | | min | max | min | max |
| Finland | FI | 8 | 5160 | 19839 | 43 | 151 |
| Sweden | SE | 10 | 5479 | 23102 | 51 | 113 |
| Norway | NO | 8 | 5163 | 20300 | 38 | 131 |
| Russia | RF | 121 | 5000 | 25000 | 30 | 123 |
| Canada | CA | 86 | 5050 | 24800 | 34 | 103 |
| United States | US | 93 | 5053 | 24786 | 33 | 139 |






**Table 2. The MARCS model setup for catchments located over the territory of the six countries**

| Country code | IRC Ave* | M1, mm yr⁻¹ | | CV | | CS/CV | PRE, mm yr⁻¹ | |
|---|---|---|---|---|---|---|---|---|
| | | Ave | Min / Max | Ave | Min / Max | Ave | Ave | Min / Max |
| FI | 1.8 | 329 | 267 / 379 | 0.21 | 0.19 / 0.24 | 0.5 | 591 | 508 / 630 |
| SE | 1.7 | 377 | 234 / 464 | 0.21 | 0.17 / 0.28 | 1.5 | 601 | 516 / 709 |
| NO | 1.1 | 446 | 372 / 590 | 0.19 | 0.17 / 0.21 | 2.5 | 459 | 374 / 601 |
| RF | 1.9 | 248 | 47.5 / 546 | 0.27 | 0.14 / 0.77 | 2.0 | 464 | 218 / 834 |
| CA | 1.9 | 344 | 50.9 / 1285 | 0.26 | 0.10 / 0.60 | 1.5 | 661 | 274 / 1909 |
| US | 2.1 | 353 | 57.5 / 1103 | 0.34 | 0.12 / 0.73 | 2.0 | 752 | 266 / 1465 |

* Ave is the value averaged over river basins located within a country





**Table 3. Mean values of annual precipitation rate (PRE$_{2050}$, mm yr$^{-1}$) calculated from the outputs of the four global climate models for 2020–2050**

| Climate projection | PRE$_{2050}$ (mm yr$^{-1}$) grouped by country | | | | | |
|---|---|---|---|---|---|---|
| | FI | SE | CA | RF | CA | US |
| CAD-26* | 662 | 687 | 532 | 538 | 723 | 805 |
| CAD-45 | 643 | 676 | 527 | 526 | 714 | 792 |
| CAD-85 | 643 | 676 | 527 | 526 | 714 | 792 |
| HAD-26 | 601 | 603 | 447 | 497 | 663 | 759 |
| HAD-45 | 601 | 615 | 489 | 502 | 691 | 748 |
| HAD-85 | 589 | 617 | 469 | 489 | 658 | 760 |
| MPI-26 | 647 | 646 | 533 | 531 | 716 | 775 |
| MPI-45 | 642 | 675 | 522 | 510 | 714 | 767 |
| MPI-85 | 678 | 656 | 539 | 528 | 722 | 759 |
| INM-45 | 609 | 620 | 465 | 480 | 692 | 763 |
| INM-85 | 617 | 653 | 502 | 498 | 715 | 736 |

* – the outputs for the CaEMS2 (CAD), HadGEM2-ES (HAD), MPI-ES-LR (MPI) and INMCM4 (INM) climate models under RCP2.6 (-26), RCP4.5 (-45) and RCP8.5 (-85) scenarios.



**Table 4. The projected mean values of ARR ($M1_{2050}$) simulated by the MARCS hydrological model under the forcing of the CaEMS2, HadGEM2-ES, MPI-EMS-LM and INMCM4 climate models**

| Climate model | Scenario | $M1_{2050}$ (mm yr$^{-1}$) by country | | | | | |
|---|---|---|---|---|---|---|---|
| | | FI | SE | NO | RF | CA | US |
| CaEMS2 | RCP26 | 367 | 432 | 516 | 288 | 377 | 377 |
| | RCP45 | 357 | 422 | 512 | 282 | 373 | 371 |
| | RCP85 | 357 | 422 | 512 | 282 | 373 | 371 |
| MPI-EMS-LM | RCP26 | 360 | 409 | 517 | 285 | 374 | 364 |
| | RCP45 | 358 | 426 | 507 | 272 | 374 | 362 |
| | RCP85 | 377 | 416 | 523 | 284 | 379 | 357 |
| $M1_{2050}$: "wet" forcing | | **363** | **421** | **515** | **282** | **375** | **367** |
| HadGEM2-ES | RCP26 | 334 | 378 | 435 | 267 | 349 | 359 |
| | RCP45 | 336 | 390 | 475 | 267 | 362 | 355 |
| | RCP85 | 328 | 386 | 457 | 261 | 345 | 361 |
| INMCM4 | RCP45 | 338 | 391 | 452 | 257 | 362 | 362 |
| | RCP85 | 343 | 409 | 488 | 266 | 374 | 348 |
| $M1_{2050}$: "dry" forcing | | **336** | **391** | **461** | **264** | **358** | **357** |
| M1: after (Rogdestvenskiy and Chebotarev, 1974) for the reference period | | **329** | **377** | **446** | **248** | **344** | **353** |





**Table 5. The coefficients of variation of ARR ($CV_{2050}$) averaged over the catchments selected**

| Climate model | Scenario | $CV_{2050}$ by country | | | | | |
|---|---|---|---|---|---|---|---|
| | | FI | SE | NO | RF | CA | US |
| CaEMS2 | RCP26 | 0.19 | 0.19 | 0.18 | 0.25 | 0.25 | 0.33 |
| | RCP45 | 0.19 | 0.19 | 0.18 | 0.25 | 0.25 | 0.33 |
| | RCP85 | 0.19 | 0.19 | 0.18 | 0.25 | 0.25 | 0.33 |
| MPI-EMS-LM | RCP26 | 0.19 | 0.20 | 0.18 | 0.25 | 0.25 | 0.33 |
| | RCP45 | 0.19 | 0.19 | 0.18 | 0.25 | 0.25 | 0.34 |
| | RCP85 | 0.19 | 0.20 | 0.18 | 0.25 | 0.25 | 0.34 |
| $CV_{2050}$ : average wet forcing | | **0.19** | **0.20** | **0.18** | **0.25** | **0.25** | **0.33** |
| HadGEM2-ES | RCP26 | 0.21 | 0.21 | 0.19 | 0.26 | 0.27 | 0.34 |
| | RCP45 | 0.21 | 0.21 | 0.18 | 0.26 | 0.26 | 0.35 |
| | RCP85 | 0.21 | 0.20 | 0.19 | 0.26 | 0.27 | 0.34 |
| INMCM4 | RCP45 | 0.21 | 0.21 | 0.19 | 0.26 | 0.26 | 0.34 |
| | RCP85 | 0.20 | 0.20 | 0.18 | 0.26 | 0.25 | 0.35 |
| $CV_{2050}$ : average dry forcing | | **0.21** | **0.20** | **0.19** | **0.26** | **0.26** | **0.34** |
| CV: after (Rogdestvenskiy and Chebotarev, 1974) for the reference period | | **0.21** | **0.21** | **0.19** | **0.27** | **0.27** | **0.34** |



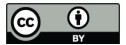

**Table 6. The ordinates of Pt3 distribution (k10/k90) and annual runoff rate (ARR10/ARR90, mm year$^{-1}$) for low (10%) and high (90%) exceedance probabilities under wet and dry forcing: the aggregation on country level**

| Time period | Values | FI | SE | NO | RU | CA | US |
|---|---|---|---|---|---|---|---|
| Reference | k10 | 1.270 | 1.270 | 1.247 | 1.350 | 1.350 | 1.446 |
| | k90 | 0.730 | 0.730 | 0.759 | 0.654 | 0.654 | 0.575 |
| | ARR10 | 418 | 479 | 556 | 335 | 464 | 510 |
| | ARR90 | 240 | 275 | 339 | 162 | 225 | 203 |
| Projected with wet forcing | $k10_{2050}$ | 1.247 | 1.260 | 1.234 | 1.325 | 1.325 | 1.432 |
| | $k90_{2050}$ | 0.759 | 0.744 | 0.770 | 0.680 | 0.680 | 0.575 |
| | $ARR10_{2050}$ | 413 | 485 | 561 | 341 | 463 | 511 |
| | $ARR90_{2050}$ | 251 | 286 | 350 | 175 | 237 | 205 |
| Projected with dry forcing | $k10_{2050}$ | 1.270 | 1.260 | 1.247 | 1.338 | 1.338 | 1.446 |
| | $k90_{2050}$ | 0.730 | 0.744 | 0.759 | 0.667 | 0.667 | 0.575 |
| | $ARR10_{2050}$ | 426 | 492 | 575 | 353 | 480 | 516 |
| | $ARR90_{2050}$ | 245 | 291 | 350 | 176 | 239 | 205 |





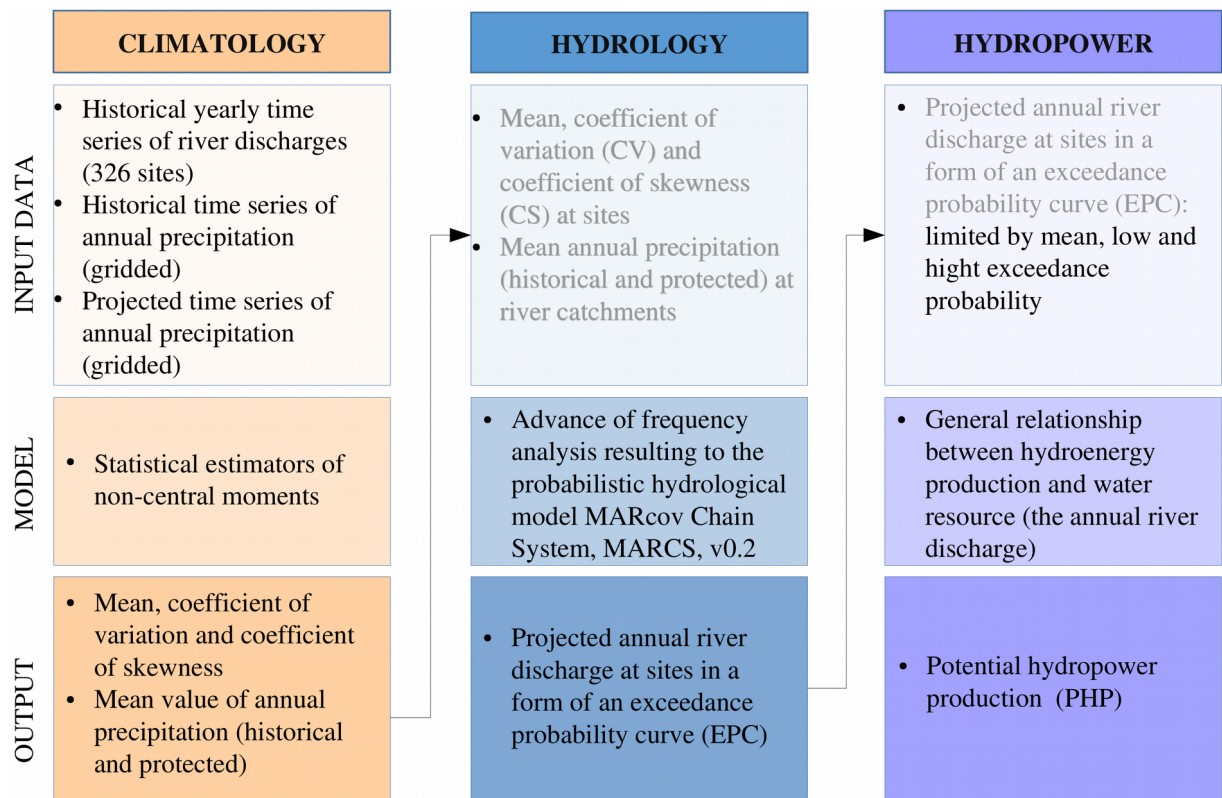

**Figure 1: A schematic presentation of the study method.**






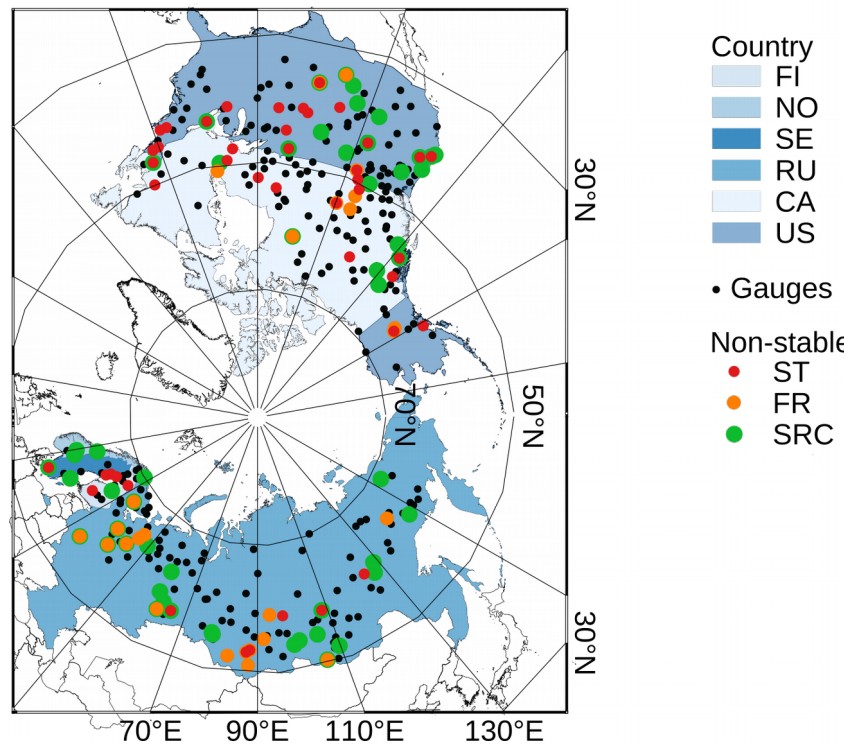

**Figure 2. Location of the catchments studied. The black dots indicate gauges with homogeneous time series, whereas the**
**red/orange dots indicate gauges with the statistically significant changes in the mean or variance (based on the Student t-test (ST) and Fisher's F-test (FR)), and the green dots mark the gauges with significant trends in observed time series (based on the Spearman Rank-Correlation Test (SRC)).**





**Figure 3. The MARCS model set-up for the Arctic for the reference period: a) the inverse of runoff coefficient (IRC); b) the mean value of annual precipitation rate (PRE, mm yr$^{-1}$); c) the second non-central moment estimate (M2, mm$^2$ yr$^{-2}$); d) the first non-central moment estimate, mean annual runoff rate (M1, mm yr$^{-1}$).**





**Figure 4. Projected changes in mean annual precipitation (dPRE$_{2050}$, %) at the catchments selected**




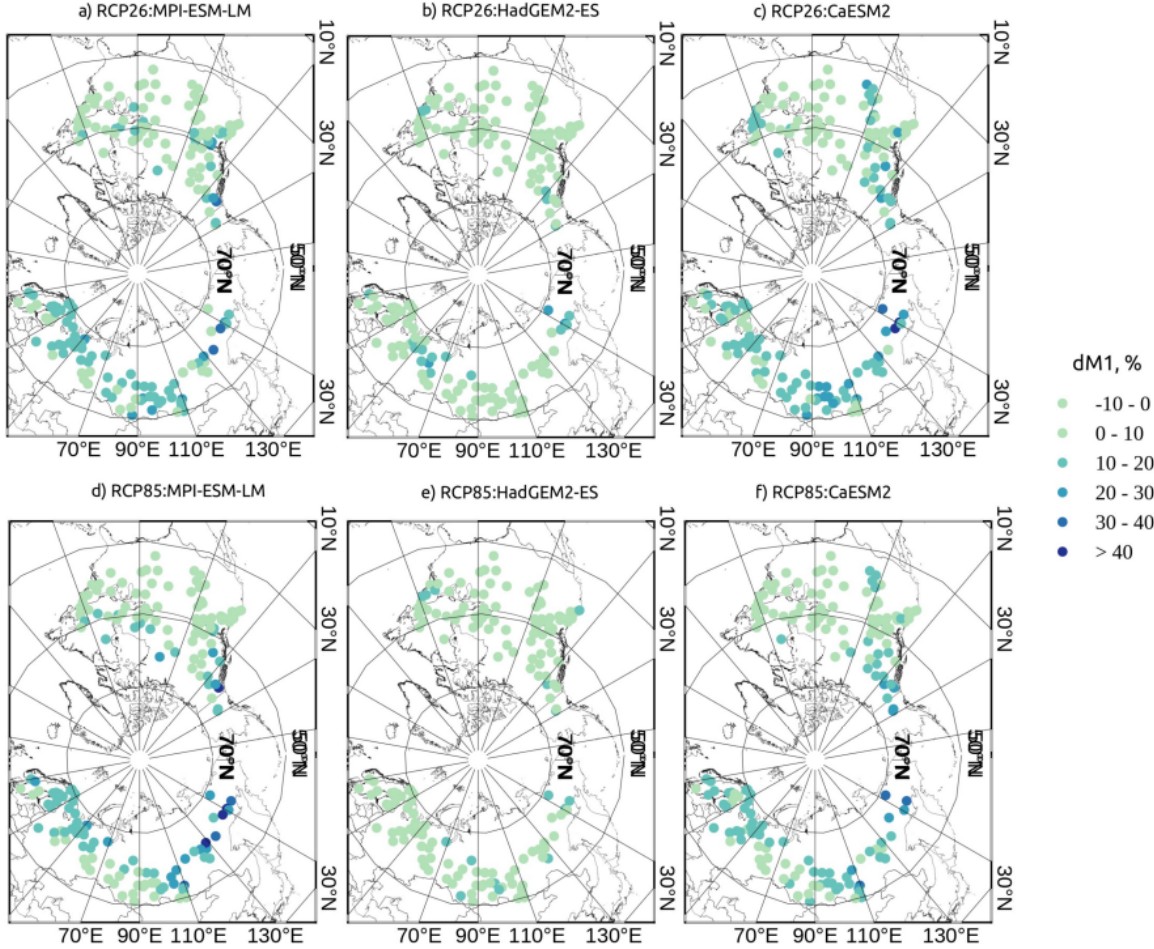

**Figure 5. Relative changes of the projected means of ARR (dM1$_{2050}$, %) for the RCP26 (top) and RCP85 (bottom)**





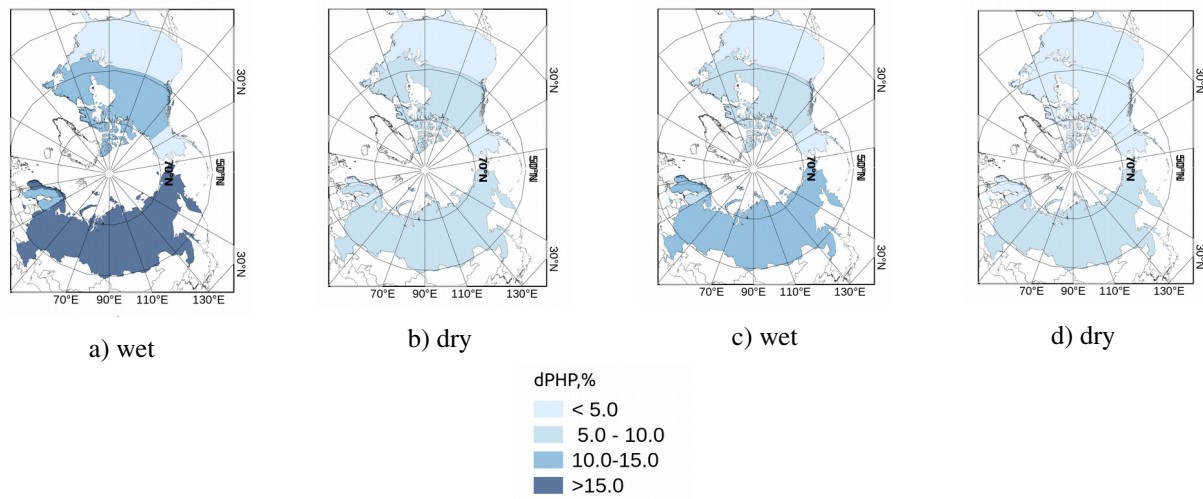

Figure 6. Relative change of the potential hydropower production (dPHP$_{2050}$) in six Arctic Council member countries according to dARR90$_{2050}$ (a, b) and dARR10$_{2050}$ (c, d) under the wet and dry forcing