# Peer review of "Climate change will increase potential hydropower production in six Arctic Council member countries based on probabilistic hydrological projections"

_Hydrology and Earth System Sciences, 2018_

## Referee Comment (RC1) · Anonymous Referee #1 · 20 Oct 2018

Shevnina et al. present an analysis of climate change impacts on hydropower production in member countries of the Arctic Council. An unusual statistical approach is applied, where moments of precipitation (computed from climate model projections) are used to project the distribution of annual river flow, which is then assumed to scale linearly with potential hydropower production. I recommend that the paper is rejected, for the following reasons:

1. Various methodological problems, including:

[Figure]

- Very confusing method, with no validation to demonstrate correct capture of annual flow or hydropower variability. There are country-level annual hydropower generation data available through EIA that ought to be used to check for correct representation of generation.

- No proof offered to show that the MARCS model simulates statistical moments of annual runoff correctly.

- No apparent filtering for catchments or reaches of river that are actually developed for hydropower already or are suitable for hydropower plants.

- No analysis or discussion as to whether the climate models used are able to provide any useful information on extreme precipitation conditions.

2. The study is behind the curve. A study published six years ago is taken to be "state-of-the-art" (line 83). In fact, there are now dozens of studies in the literature that examine hydropower production under future climate conditions, including monthly simulations of individual plants at global and regional scales. You need to build from the most up-to-date work in the field to demonstrate your contribution. If your method offers something that can't be achieved with the existing tools, then you need to demonstrate the advantages and performance.

3. Very little in the way of new knowledge in the conclusions. There are various published studies that show wetter climate and increased hydropower generation in northern latitudes.
* * *

---

## Referee Comment (RC2) · Anonymous Referee #2 · 20 Oct 2018

Review of HESS-2018-473

The paper deals with a method to estimate hydropower potential for six Arctic countries. This is interesting in the current debate on renewable energy, energy storage and balancing non-storable sources. And as such this could be a relevant paper, but I do think it needs significant clarifications and improvements in the description of the methods, the results and not at least in the discussion of the findings. Some major issues:

- The method proposed lack a proper demonstration of its applicability to the current conditions. There are no data that shows that the hydrology or production under current conditions are properly reproduced. I do not think the description of the model was particularly easy to follow either.

- In the computation of the hydropower production, how is the head estimated? Particularly for countries with large high head systems this would be important to know.

- To what extent do current regulations influence output from the model? It seems that e.g. the Norwegian data used are heavily influenced by current regulations. What bias can this lead to and is this taken care of in the analysis?

- How is the baseline for the production used in generating the results presented e.g. in figure 6 estimated? How well does this baseline values correspond with known production? Data are available from the energy agency and from literature (e.g. Hoes et al. (2017) PLOS One). Were there any corrections done to get this right in the current analysis?

- The hydropower output is only presented as an aggregated value in figure 6. I do miss some more detail on the results leading up to this, particularly since this is the topic of the paper.

- The discussion sections tend to rather discuss the MARCS output and discharge and precipitation data rather than hydro power and energy production which is the topic of the paper.

- There is a number of hydropower studies available in literature, and some is cited in the manuscript, and the authors state that their contribution is a better assessment of variability and uncertainty of the future predictions. This is interesting, but unfortunately not much discussed in the manuscript. How does your predictions with better assessment of variability compare to previous studies? Generally, I think the discussion section lack a proper discussion of the findings of this paper in relation to what is available in literature and how the results of this paper relate to previous findings.

- There is a body of literature on this topic available, but some important recent work is missing in the current manuscript:
    - van Vliet et al. (2016) Nature Climate Change

- van Vliet et al. (2016) Global Environmental Change
- Flörke et al. (2012) J.Water Clim.Change
- A number of regional and single system studies exists, also in the region studied in this manuscript

I do think these should be discussed in relation to the method and findings in this manuscript, see also comment above. Based on this discussion, what is the major benefit of the proposed method and what new insight does it provide? As stated before, you say there is a benefit in your way of doing the assessment of hydropower potential, but you do not present a convincing argument that this is the case in the paper.

- In the discussion it is stated that the results have the highest potential for use where there is new hydro power planned. I am not sure I agree, since altered inflow will greatly influence existing plants regarding operational changes, possible expansions and upgrading (which is important topics in the hydropower industry).

- Looking at the results, not only volume is important but also seasonal distribution of water. The timing of the extra inflow might be as important as the percentage increase, and to increase the relevance of the paper this is a topic that should be addressed.

- P2-l61: Is the discussion on water-stress indicators at all relevant to this study?

---

## Referee Comment (RC3) · Anonymous Referee #3 · 21 Oct 2018

In this study, the authors used projected precipitation from 4 GCMs to estimate the likely future streamflow using MARCS model. They then used a simple PHP formula to discuss how the projected change of streamflow may affect potential hydropower production. They concluded that climate change will increase potential hydropower production in six Arctic Council member countries.

Overall, while there is interest to understand this topic, I don't think there are sufficient data and appropriate methods to support this assessment. My detailed concerns are provided below. Given the current state of this manuscript, I would regretfully recom-

[Figure]

**HESSD**

mend decline this manuscript from further publication in HESS.

1. [Insufficient GCM Representation] If the focus is to identify the most credible projection of future water resources (for hydropower production and other uses), individual runs from 4 selected GCMs are obviously insufficient. During the period of 2020–2050, the main controlling factor is the interannual variability of precipitation (modeled by different GCMs, as well as the ensemble simulations modeled by one single GCM with a series of perturbed initial conditions). With the large interannual variability, a much larger set of GCM projections should be used to capture the uncertainty. As a matter of fact, given the simplicity of MARCS and the selected PHP approach, I see no reason why the authors couldn't and shouldn't use more (if not all) existing CMIP5 results to conduct their analysis and draw more defensible conclusion. With that said, currently I don't think there is sufficient GCM projections to support the assessment and findings of this study.

2. [Treatment of Precipitation] I think some gauges examined by the authors are too large for grid-based assessment (i.e., contributing watershed covering multiple GCM grids). In such cases, using a single grid precipitation to represent the total precipitation input to the watershed is inappropriate and biased. With the advance of GIS techniques and data in the recent decade, I believe the authors can use the watershed boundary of each selected site as a spatial filter to more appropriately summarize average precipitation into the watersheds. This can hopefully help reduce some erroneous Q>P cases (i.e., total volume of streamflow is greater than precipitation) reported by the authors in the current manuscript.

3. [Limitations of MARCS and the Overall Statistical Approach] While I see the value of MARCS to potentially support large ensemble assessment (given its simplicity), I suspect if it is a suitable approach to explore future water availability in the context of atmospheric warming and climate change. In essence, the current assessment used statistical approach to estimate the likely change of streamflow ONLY by the change of precipitation (projected by GCMs). This approach would neglect other temperature-

related nonlinear effects such as earlier snowmelt and enhanced evapotranspiration.

4. [Oversimplification of Hydropower Assessment] While hydropower was specifically called out in the title, the authors only used a very simplified model (PHP) to infer the likely influence of climate change on hydropower production only from water availability perspective. Yes, for the selected region, the overall runoff is likely to increase as the result of increasing precipitation suggested by many GCMs, but the increases are likely in forms of more severe hydrologic extremes. With the intensified hydrologic extreme events, will our current reservoirs have sufficient storage to accommodate these highly varied inflows and be able to operate in the same fashion? For run-of-river types of hydroelectric projects, they may end up spilling most of the increased runoff due to limited storage so won't see a corresponding increase in hydropower generation. These more pressing issues cannot be addressed through an over-simplified hydropower model selected by the authors.

5. [Bias-correction Method] The delta bias-correction approach has become an outdated method. The authors should at-least consider the quantile-based bias-correction approach that can better adjust the GCM biases.

6. [Regulated Streamflow] I suspected that many of the gauge data used by the authors are in-fact regulated by existing reservoirs. How will this affect your assessment?

Other minor concerns:

7. [Line 27] "Engineering hydrology" sounds awkward.

8. [Line 186] Are you referring the "University of Delaware Air Temperature & Precipitation"? If so, a proper reference and citation should be provided.

9. [Lines 212–213] I don't think the statement is true. In the U.S., the cumulative streamflow originated from inland is far more important than the near-coast water resources for the purpose of hydropower generation.

10. [Lines 216 and 217] Should be Figure 3d.

11. [Line 218] Should be Figure 3c.

12. [Line 236] Should be Table 3.

13. [Line 242] Should be Figure 4.

14. [Line 282] Typo – "ew".

15. [Table 3] The first CA should be NO? Also, please provide the unique CMIP5 ensemble number (e.g., r1p1i1) so that the authors may know which exact GCM runs were used.

16. [Figure 1] In the bottom left box, "protected" should be "projected"?

17. [Figure 2] It is very difficult to differentiate sites with overlapping ST, FR, or SRC.

---

## Author Comment (AC1) · 7 Dec 2018

The Referee #1 recommends to reject the manuscript due to "the following reasons:

1. Various methodological problems, including: - Very confusing method, with no validation to demonstrate correct capture of annual flow or hydropower variability. There are country-level annual hydropower generation data available through EIA that ought to be used to check for correct representation of generation." Answer: In this particular manuscript we did not pay much attention to the method itself since there are

more papers to describe the details (Shevnina and Silaev, 2018; Shevnina et al., 2017; Kovalenko, 2014; Viktorova and Gromova, 2008; Kovalenko, 1993). This method is suggested more than 20 years ago, and it is included to the course "Hydrological modeling" for Ms. and PhD students of the Russian State Hydrometeorological University. Most studies in the Fokker-Plank-Kolmogorov approach in hydrological modelling are published in Russian and there are also the manuscripts in English listed in the section of References. The discussion about the validation of the method is given further. As for the hydropower generation data, we think there is a typo, and the Referee #1 refers to IEA (https://www.iea.org/) instead of EIA (U.S. Energy Information Administration). Nevertheless, the IEA has data of hydropower generation per county, but we are not sure if that is freely available, and we do not estimate the potential hydropower production on an absolute value.

"- No proof offered to show that the MARCS model simulates statistical moments of annual runoff correctly. Answer: The model cross-validation procedure is described by Shevnina et al., (2017) in English and by Kovalenko (1993) in Russian. It includes several steps: to define the sub-periods in the observed yearly time series of river discharges; to set-up the model for the first sub-period; to simulate the exceedance probability curve (EPC) for the second sub-period and to compare with the EPC constructed from observations in the second sub-period (Kovalenko, 1993). The results on the cross-validation procedure for the annual runoff are already published by Kovalenko (1993) based on the historical observations on the catchments located in the North of Russia. In this study, we relay on these results and this circumstance was mentioned on the page 7 lines 210-211.

"- No apparent filtering for catchments or reaches of river that are actually developed for hydropower already or are suitable for hydropower plants." Answer: The data on the observed yearly time series of river discharges was filtered by applying the statistical tests to reveal non-homogeneity/trends in the time series according to Dahmen and Hall (1990), and to calculate the length of the reference period. In this case, the current
regulation rules can affect on the projected statistical moments of the annual runoff for the catchments with present hydropower network. To answer to the questions How? or How much?, a new study is needed.

"- No analysis or discussion as to whether the climate models used are able to provide any useful information on extreme precipitation conditions." Answer: In this study, we did not discuss the ability of the climate models to provide the projected values of the extreme precipitation conditions because the basic parameterization scheme by Kovalenko (1883) was applied. It means that the projected variance of precipitation was not accounted in the calculations, and the statistical moments of annual runoff were simulated only based on the mean of annual precipitation. To account the projected extremes of precipitation the method should be modified, and this is the next step further. This step can be done as soon as the simplest version of the AFA method (https://www.geosci-model-dev-discuss.net/gmd-2018-108/) will be discussed and published.

2. The study is behind the curve. A study published six years ago is taken to be "state-of-the-art" (line 83). In fact, there are now dozens of studies in the literature that examine hydropower production under future climate conditions, including monthly simulations of individual plants at global and regional scales. You need to build from the most up-to-date work in the field to demonstrate your contribution. If your method offers something that can't be achieved with the existing tools, then you need to demonstrate the advantages and performance. Answer: We can update the "state of the art" hydropower production literature, but it would indeed be nice to get clarification if the problem is with the method or with the lack of references to recent studies. We are not sure that the method itself is clear for the hydrological modeling community, where physically-based hydrological models are most common. These physically-based hydrological models are well developed tools to serve a short-term forecasting, and in many studies they are used to simulate "climate based" projections of streamflow runoff.

3. Very little in the way of new knowledge in the conclusions. There are various published studies that show wetter climate and increased hydropower generation in northern latitudes. Answer: Yes, we agree that more discussion should be given to the novelty of the study in terms of usefulness of the presented method to planning of hydropower production. In the revised version, we can expand the discussion section based on the papers like this: https://www.hydrol-earth-syst-sci.net/21/133/2017/hess-21-133-2017.html.

---

## Author Comment (AC2) · 7 Dec 2018

The Referee #2 recommends to do "significant clarifications and improvements in the description of the methods, the results and not at least in the discussion of the findings." The following major issues were mentioned:

"The method proposed lack a proper demonstration of its applicability to the current conditions. There are no data that shows that the hydrology or production under current conditions are properly reproduced. I do not think the description of the model was

particularly easy to follow either."

Answer: We agree, that only proper comparison to observations proves modeling results. The method used in this study, also offers the comparison of the observed and modeled probability distribution functions (see page 8 in the Supplement). The specific cross-validation procedure includes several steps: to define the sub-periods in the observed yearly time series of river discharges; to set-up the model for the first sub-period; to simulate the exceedance probability curve (EPC) for the second sub-period and to compare with the EPC constructed from observations in the second sub-period (Kovalenko, 1993). The results on the cross-validation procedure for the annual runoff are already published by Kovalenko (1993) based on the historical observations on the catchments located in the North of Russia. In this study, we relay on these results and this circumstance was mentioned on the page 7 lines 210-211. It should be noted, that in this particular manuscript we did not pay much attention to the method itself since there are more papers to describe the details (Shevnina and Silaev, 2018; Shevnina et al., 2017; Kovalenko, 2014; Viktorova and Gromova, 2008; Kovalenko, 1993). This method developed more then 20 years ago, and even there are plenty papers published in Russian, but only few manuscripts are available in English. Kovalenko, 2014: Russ. Meteorol. Hydrol. 39:115. doi:10.3103/S1068373914020071. Shevnina and Silaev, 2018: GMD. doi: 10.5194/gmd-2018-108. Viktorova and Gromova, 2008: Russ. Meteorol. Hydrol. 33:6. doi: 10.3103/S1068373908060071.

"In the computation of the hydropower production, how is the head estimated? Particularly for countries with large high head systems this would be important to know." Answer: We agree, that the head system is needed to evaluate the potential hydropower production (PHP) since the Eq. (1) was applied in this study. However, the PHP was not evaluated on absolute values even on the country level, which was finally done in this study. In this study, we have shown the relative changes in the water resource in terms of probability, i.e for the annual runoff of 10/90 % of exceedance probability and it was assumed that these changes are linearly related to the the PHP in all range of

the EPC (see p. 12, the Supplement). To estimate the PHP in absolute values, the information on the site-specific head system is needed. It could be the topic for "a case study" manuscript in the future.

"To what extent do current regulations influence output from the model? It seems that e.g. the Norwegian data used are heavily influenced by current regulations. What bias can this lead to and is this taken care of in the analysis?" Answer: In fact, the data on the observed yearly time series of river discharges was filtered only formally by applying the statistical tests to reveal non-homogeneity/trends in the time series and to calculate the length of the reference period. In this case, the current regulation rules can affect on the projected statistical moments of the annual runoff for the catchments with present hydropower network. To answer to the question: how?, a new study would be required. However, to revise this study we added the information on the current regulations for the catchments (e.g. Norway), which were chosen to model set-up, and we discussed the possible effects as well.

"How is the baseline for the production used in generating the results presented e.g. in figure 6 estimated? How well does this baseline values correspond with known production? Data are available from the energy agency and from literature (e.g. Hoes et al. (2017) PLOS One). Were there any corrections done to get this right in the current analysis?" Answer: In this study, the only relative changes on the potential hydropower generation were suggested based on an assumption that they are simply related to the changes in the annual runoff in all range of the exceedance probability. Thus the only estimations for the annual runoff for the baseline were estimated from the river runoff observations to compare with the projections of the annual runoff. This circumstance was mentioned in the section 2.1.3, but may not clearly. It means, that the relative changes is the hydropower production in the Fig. 6 actually show the relative changes in the annual runoff of low and high exeedance probability. Recently, we used another way to present the results of our study (see section 2.1.3 and Fig. 6 in new version of the manuscript). It is also possible to evaluate the PHP in absolute values

based on the approach used in Hoes et al. (2017) can be done in the near future.

"The hydropower output is only presented as an aggregated value in figure 6. I do miss some more detail on the results leading up to this, particularly since this is the topic of the paper." Answer: Recently, the results of our estimation were aggregated on the country level since the detailed analysis comes to be a topic of "a case study" on a country level (at least for Finland). In this case, the details on the changes on the hydropower production on "a river catchment" level will be provided. During the revising, we realized that the title of the paper actually does not fit the content of the manuscript, which is more about the probabilistic projections of the annual runoff.

"The discussion sections tend to rather discuss the MARCS output and discharge and precipitation data rather than hydro power and energy production which is the topic of the paper." Answer: We agree, that recently the title of the manuscript does not correctly express the idea, thus it should be changed. Even though the title would change, we think more details would be provided in the revised version of the manuscript.

"There is a number of hydropower studies available in literature, and some is cited in the manuscript, and the authors state that their contribution is a better assessment of variability and uncertainty of the future predictions. This is interesting, but unfortunately not much discussed in the manuscript. How does your predictions with better assessment of variability compare to previous studies? Generally, I think the discussion section lack a proper discussion of the findings of this paper in relation to what is available in literature and how the results of this paper relate to previous findings." Answer: We agree, the contribution of this study was poorly discussed and the additional comparisons with recent studies was done in the revised version.

"There is a body of literature on this topic available, but some important recent work is missing in the current manuscript: van Vliet et al. (2016) Nature Climate Change; van Vliet et al. (2016) Global Environmental Change; Flörke et al. (2012) J.Water Clim.Change. A number of regional and single system studies exists, also in the re-

gion studied in this manuscript I do think these should be discussed in relation to the method and findings in this manuscript, see also comment above. Based on this discussion, what is the major benefit of the proposed method and what new insight does it provide? As stated before, you say there is a benefit in your way of doing the assessment of hydropower potential, but you do not present a convincing argument that this is the case in the paper." Answer: We agree, the contribution of this study was poorly discussed and the additional connection with recent studies (including the listed above) will be done in the revised version.

"In the discussion it is stated that the results have the highest potential for use where there is new hydro power planned. I am not sure I agree, since altered inflow will greatly influence existing plants regarding operational changes, possible expansions and upgrading (which is important topics in the hydropower industry)." Answer: In our opinion, the methods of risk analysis are should to be applied to show how to utilize the result of the study. We think that it is only a way to show the practical effect of any probabilistic forecasts, not only hydrological or meteorological. Recently, we did not find any specialist in the risks assessment to clarify the situation with the potential of the probabilistic hydrological projections in hydropower planning, however we hope that it will once happen. In our study, the probabilistic projections of the annual river runoff were presented to show the relative changes in water resources in the North. The relative changes of the water resources were simply related the potential hydropower production. We agree, that this level of aggregation is not enough to give any recommendations in i.e. an optimal operation on a particular hydropower station. However, the "catchment scale" aggregation will be next step while the perspective of the probabilistic form of long term hydrological projections in risks assessment will be clarify.

"Looking at the results, not only volume is important but also seasonal distribution of water. The timing of the extra inflow might be as important as the percentage increase, and to increase the relevance of the paper this is a topic that should be addressed.

Answer: We agree, that the seasonal distribution of water inflow to hydropower plants is more important to plan the regulation rules. The method used in this study allows evaluation also seasonal distribution of water (see e.g. Domínguez and Rivers, 2004). However, the probabilistic hydrological model should be substantially improved and it require a separate study and new model core. Recently, the simplest version of the core was applied (Shevnina et al., 2017), but even this version the approach makes issues in understanding by hydrologists get used to deal with physically-based "rainfall-runoff" models.

"P2-l61: Is the discussion on water-stress indicators at all relevant to this study?" Answer: We agree, and the discussion on water-stress indicators was removed from the revised version of the manuscript.

Please also note the supplement to this comment:
https://www.hydrol-earth-syst-sci-discuss.net/hess-2018-473/hess-2018-473-AC2-supplement.pdf

---

## Author Comment (AC3) · 7 Dec 2018

The Referee #3 recommends to "decline this manuscript from further publication in HESS." The following major issues were mentioned:

1. "[Insufficient GCM Representation]". . . If the focus is to identify the most credible projection of future water resources (for hydropower production and other uses), individual runs from 4 selected GCMs are obviously insufficient. During the period of 2020–2050, the main controlling factor is the interannual variability of precipitation (modeled by dif-

ferent GCMs, as well as the ensemble simulations modeled by one single GCM with a series of perturbed initial conditions). With the large interannual variability, a much larger set of GCM projections should be used to capture the uncertainty. As a matter of fact, given the simplicity of MARCS and the selected PHP approach, I see no reason why the authors couldn't and shouldn't use more (if not all) existing CMIP5 results to conduct their analysis and draw more defensible conclusion. With that said, currently I don't think there is sufficient GCM projections to support the assessment and findings of this study. Answer: We agree, that four selected GCMs are insufficient to represent whole spectrum of the future changes on water resources, and ensemble of the outputs may gives more sufficient estimations on a range of changes. It is a regular way to get the probabilistic form of hydrological predictions simulated by the physically based hydrological modeling. In this case, the ensemble of river discharges simulated under the ensemble of the meteorological forcing allow to estimate the river runoff of particular exceedance probability. However, in this study the method used allows to simulate the exceedance probability curves (EPC) of annual runoff already from a single climate projection (see Table 2 in the Supplement), not from an ensemble of projections. In this manuscript 11 climate projections were used to define two types for the future climate: "wet" and "dry", and to simulate the annual runoff for low/high exceedance probability. It is still not clear how to used the ensemble of EPC in long term planning in hydropower production (see discussion on p. 11). Actually, we have trying to define the perspective of the probabilistic form hydrological projections to estimate potential hydropower generation on a long-term perspective. In this study we used the simplest method to transfer water resource to the hydropower production in assumption that these two values linearly related in whole range of probability of exceedance. However, we would guess that it is not enough to show possible application in the long-term planning in hydropower production. Our opinion is that it could be possible only by involving of the methods of risk assessment. Recently, the probabilistic form of forecasts are common in operational practice, and even in this case it is difficult to utilize this form in practice. As soon as the method in how the probabilistic projections of annual runoff to estimate

potential hydropower generation will be clear, a number of GCMs used in simulations of the future water resource can be increased. The method used allows to account the interannual variability of precipitation from the climate projections, however we consider to modify the MARCS model for this case on the following study.

2. "[Treatment of Precipitation]" ... I think some gauges examined by the authors are too large for grid-based assessment (i.e., contributing watershed covering multiple GCM grids). In such cases, using a single grid precipitation to represent the total precipitation input to the watershed is inappropriate and biased. With the advance of GIS techniques and data in the recent decade, I believe the authors can use the watershed boundary of each selected site as a spatial filter to more appropriately summarize average precipitation into the watersheds. This can hopefully help reduce some erroneous Q>P cases (i.e., total volume of streamflow is greater than precipitation) reported by the authors in the current manuscript. Answer: Yes, we agree that one grid point per watershed may be insufficient to represent the mean of precipitation for the catchments of small/big sizes. It was discussed in the first paragraph on p. 11. The regional scale climate models provide an opportunity to calculate the mean of precipitation as the average over the values located within the watershed boundary, and it may improve the representation of the hydrological projections. We plan to do it in the near future. This manuscript was addressed more the application of long term probabilistic hydrological projections to estimate the potential hydropower production. While the approach may be accepted, the role of biases on precipitation can be study to define the optimal method to calculate the mean values of precipitation from projected climatology to force the probabilistic hydrological model MARCS.

3. "[Limitations of MARCS and the Overall Statistical Approach]" ... While I see the value of MARCS to potentially support large ensemble assessment (given its simplicity), I suspect if it is a suitable approach to explore future water availability in the context of atmospheric warming and climate change. In essence, the current assessment used statistical approach to estimate the likely change of streamflow ONLY by the change

of precipitation (projected by GCMs). This approach would neglect other temperature CO2 related nonlinear effects such as earlier snowmelt and enhanced evapotranspiration. Answer: The authors agree that the changes in the air temperature affect to the future water availability via enhancing evapotranspiration, and it should be accounted in the hydrological models. In the probabilistic hydrological model MARCS, the projected mean of air temperature is included to the simulations via a regional oriented parameterization scheme allowing to improve the results of the model verification (cross-validation): Shevnina et al., 2017, Kovalenko, 1993. However, in this study the basic parameterization scheme was applied, it gives of over 80 % "good" hindcasts for the simulated exceedance probability curves of annual runoff (Kovalenko, 1993). Thus, only the projected mean of precipitation was used in the simulations. To look forward, we plan to develop the regional oriented scheme for the territory of six Arctic Council members countries, and to include the changes in the air temperature in the projections of water availability. In this particular study, we would prefer to stress the features of the simplest version of the MARCS model, and to found the perspective of long-term probabilistic form of hydrological projections to be apply in evaluations of the future potential hydropower production. In this case, our opinion that the title of the manuscript does not fit the content and should be changed.

4. "[Oversimplification of Hydropower Assessment]"... While hydropower was specifically called out in the title, the authors only used a very simplified model (PHP) to infer the likely influence of climate change on hydropower production only from water availability perspective. Yes, for the selected region, the overall runoff is likely to increase as the result of increasing precipitation suggested by many GCMs, but the increases are likely in forms of more severe hydrologic extremes. With the intensified hydrologic extreme events, will our current reservoirs have sufficient storage to accommodate these highly varied inflows and be able to operate in the same fashion? For run-of-river types of hydroelectric projects, they may end up spilling most of the increased runoff due to limited storage so won't see a corresponding increase in hydropower generation. These more pressing issues cannot be addressed through an over-simplified hydropower model selected by the authors. Answer: We agree that the method to transfer the water resource to the economic value (the hydropower production) is very simple, and it needs only for the mean value of annual runoff, the site-specific head and the technology-specific constant (see the Eq. 1). Moreover, the very simple assumption that the changes on the annual runoff is simply related to the changes in the potential hydropower production in whole range of exceedance probabilities. However, these assumptions allow to evaluate the potential hydropower production in terms of probability, and to "look up" the risks connected to the annual runoff of low and high exceedance probability (or percentiles of annual runoff in terms of the statistic). In our opinion, the methods of a risk assessment should be applied to transfer the water resource to the hydropower production in the future study. In this case, it would be possible to answer questions as "will our current reservoirs have sufficient storage to accommodate these highly varied inflows and be able to operate in the same fashion?" or how to account the side-specific information on the hydropower type, head and operational rules. This study is focused more in the features of the probabilistic hydrological projections, and show the global scale assessment on the potential hydropower production. The study provided more details on the method applied to simulate the probabilistic projections of annual runoff, and the title is about the hydropower production. In our opinion, the title of the manuscript should be changed.

5. "[Bias-correction Method]"... The delta bias-correction approach has become an outdated method. The authors should at-least consider the quantile-based bias-correction approach that can better adjust the GCM biases. Answer: It is not clear what does it mean that the "bias-correction approach has become an outdated method"? Does it means that this method gives significant errors to be applied to correct the forcing for the hydrological physically-based, balance or probabilistic models? In our opinion, a simple method is preferable if it gives results similar to a complex method. Thus, it is important to answer the question: how the method of climate correction affects to the results simulated by the probabilistic hydrological model MARCS. In this study we started from the simplest version of climate correction, and in the future, more

sophisticated methods of climate correction can be applied to study the sensitivity of the MARCS model to biases in the climate forcing. We added this circumstance in the section of Discussion in the revised version of the manuscript.

6. "[Regulated Streamflow]" . . . I suspected that many of the gauge data used by the authors are in-fact regulated by existing reservoirs. How will this affect your assessment? Answer: In fact, the existing regulation by reservoirs may affect to the yearly values of observed river discharges only in case if there is a multi-year re-distribution of water inflow. The "regulated" gauges located for example on the territory of Norway operate on seasonal redistribution of water inflow, thus the regulation rules are not affect to the yearly discharges. In our study, the observed yearly time series of river discharges were filtered by applying the statistical tests to reveal non-homogeneity/trends in the time series and to calculate the length of the reference period. However, the current regulation rules can affect on the projected statistical moments of the annual runoff for the catchments with present hydropower network. To answer to the question: how?, a new study would be required. In this study we added the information on the current regulations for the catchments (e.g. Norway), which were chosen to model set-up, and we discussed the possible effects as well.

Other minor concerns were also mentioned by the Referee: 7. "[Line 27] "Engineering hydrology" sounds awkward." Answer: This term was replace by the "environmental engineering", and comments were added to specify this branch of the hydrology withing the environmental science.

8. "[Line 186] Are you referring the "University of Delaware Air Temperature & Precipitation"? If so, a proper reference and citation should be provided." Answer: The proper reference is now included into the text.

9. "[Lines 212–213] I don't think the statement is true. In the U.S., the cumulative streamflow originated from inland is far more important than the near-coast water resources for the purpose of hydropower generation." Answer: The statement is removed.

10. "[Lines 216 and 217] Should be Figure 3d" Answer: we agree.

11. [Line 218] Should be Figure 3c. Answer: we agree.

12. [Line 236] Should be Table 3. Answer: we agree.

13. [Line 242] Should be Figure 4. Answer: we agree.

14. [Line 282] Typo – "ew". Answer: we correct the typing mistake.

15. [Table 3] The first CA should be NO? Also, please provide the unique CMIP5 ensemble number (e.g., r1p1i1) so that the authors may know which exact GCM runs were used. Answer: We add this information into the text.

16. [Figure 1] In the bottom left box, "protected" should be "projected"? Answer: we correct the typing mistake.

17. [Figure 2] It is very difficult to differentiate sites with overlapping ST, FR, or SRC. Answer: we correct the figure.

Please also note the supplement to this comment:
https://www.hydrol-earth-syst-sci-discuss.net/hess-2018-473/hess-2018-473-AC3-supplement.pdf